# Molecular classification of the placebo effect in nausea

Karin Meissner[1,2]☉*, Dominik Lutter[3,4]☉, Christine von Toerne[5], Anja Haile[1], Stephen C. Woods[6], Verena Hoffmann[1], Uli Ohmayer[5], Stefanie M. Hauck[5‡], Matthias H. Tschoep[3,4,7‡]

1 Institute of Medical Psychology, Faculty of Medicine, LMU Munich, Munich, Germany, 2 Division of Health Promotion, Coburg University of Applied Sciences, Coburg, Germany, 3 Institute of Diabetes and Obesity, Helmholtz Diabetes Center, Helmholtz Zentrum München, German Research Center for Environmental Health (GmbH), Neuherberg, Germany, 4 German Center for Diabetes Research (DZD), Neuherberg, Germany, 5 Research Unit Protein Science, Helmholtz Zentrum München, German Research Center for Environmental Health (GmbH), Neuherberg, Germany, 6 Department of Psychiatry and Behavioral Neuroscience, Metabolic Diseases Institute, University of Cincinnati, Cincinnati, Ohio, United States of America, 7 Division of Metabolic Diseases, Department of Medicine, Technische Universität München, Munich, Germany

☉ These authors contributed equally to this work.
‡ These authors also contributed equally to this work.
* karin.meissner@med.lmu.de

**Data Availability Statement:** The mass spectrometry proteomics data have been deposited to the ProteomeXchange Consortium via the PRIDE partner repository with the dataset identifier

## Abstract

In this proof-of-concept study, we tested whether placebo effects can be monitored and predicted by plasma proteins. In a randomized controlled design, 90 participants were exposed to a nauseating stimulus on two separate days and were randomly allocated to placebo treatment or no treatment on the second day. Significant placebo effects on nausea, motion sickness, and (in females) gastric activity could be verified. Using label-free tandem mass spectrometry, 74 differentially regulated proteins were identified as correlates of the placebo effect. Gene ontology (GO) enrichment analyses identified acute-phase proteins and micro-inflammatory proteins to be involved, and the identified GO signatures predicted day-adjusted scores of nausea indices in the placebo group. We also performed GO enrichment analyses of specific plasma proteins predictable by the experimental factors or their interactions and identified 'grooming behavior' as a prominent hit. Finally, Receiver Operator Characteristics (ROC) allowed to identify plasma proteins differentiating placebo responders from non-responders, comprising immunoglobulins and proteins involved in oxidation reduction processes and complement activation. Plasma proteomics is a promising tool to identify molecular correlates and predictors of the placebo effect in humans.

## Introduction

The neurobiological mechanisms underlying placebo effects have become a research topic of increasing academic interest and intense study over the last decade. Approaches toward identification of exact mechanistic underpinnings frequently focus on changes in brain activity and

PXD020563.(https://www.ebi.ac.uk/pride/help/archive/reviewers).

**Funding:** KM (ME3675/1-1), DFG Research Unit FOR 1328, German Research Foundation, https://www.dfg.de/en/. The funder played no role in the study design, data collection and analysis, decision to publish, or preparation of the manuscript.

**Competing interests:** The authors have declared that no competing interests exist.

brain connectivity to the release of neurotransmitters, including endogenous opioids, endo-cannabinoids, and dopamine [1, 2]. Few studies focused on the peripheral mechanisms of expectancy-induced placebo effects and revealed autonomic changes at the end-organ level, for example in gastric activity and coronary perfusion [3–7].

For the field of clinical research, however, a more profitable approach would be the discovery of circulating biomarkers, which would provide the potential to monitor placebo effects in clinical trials without having to include cumbersome, expensive and invasive placebo control groups. To date, few studies have sought accessible markers for the placebo effect in blood, and they typically followed a narrow candidate-based approach. In a randomized controlled trial on irritable bowel syndrome [8], for example, patients in the placebo group showed a more pronounced reduction in serum osteoprotegerin levels than waitlist controls. Furthermore, placebo-treated patients with satisfactory relief ('placebo responders') had higher baseline levels of osteoprotegerin and TWEAK than placebo non-responders. These results underscore the potential of plasma proteins to monitor and predict placebo effects in clinical trials.

Recent advances in high-throughput proteomics in combination with next-generation bioinformatics have now enabled novel ways to identify the molecular fingerprint of placebo effects in human plasma. Blood plasma is the most sampled and most complex human proteome and comprises immunoglobulins, peptide and protein hormones, proteins secreted by solid tissues (e.g., liver proteins), cytokines, lysosomal proteins, tissue leakage proteins (e.g. troponin, creatine kinase), and proteins released from tumors and infectious organisms. Depending on their site of origin, plasma proteins can change within minutes to hours [9]. Not only single candidates but also protein signatures and associated biological processes can be identified by proteomic approaches. Plasma proteomes can be quantitatively compared between two or more samples, thereby enabling differential protein expression analyses, both between groups and over time [10].

Recently, our team completed a randomized controlled trial in healthy volunteers that demonstrated significant placebo effects on symptom ratings in experimentally-induced nausea [11]. Expanding the findings of this study, we here aimed to identify proteins and associated biological processes that have the potential to track the placebo effect in peripheral plasma. Furthermore, we aimed to identify plasma proteins that could predict 'placebo responders', i.e. participants in the placebo groups with adequate symptom relief.

## Materials and methods

### Study design

In this randomized controlled trial, 100 healthy participants were exposed to a virtual vection drum on two separate days. On Day 2, participants were randomly assigned to either placebo treatment (i.e., sham stimulation of a sham acupuncture point; n = 60), or active treatment (i.e., transcutaneous electrical nerve stimulation (TENS) of the acupuncture point 'PC6'; n = 10), or to no treatment (n = 30). The active treatment group (data not analyzed) was included to allow for the blinded administration of the placebo intervention, a common approach in placebo studies [12]. The no treatment group served to control the placebo effect for naturally occurring changes from Day 1 to Day 2. Placebo and control groups were stratified by gender (Fig 1A). Computer-assisted randomization was performed by a person not involved in the experiments, who prepared sequentially numbered, sealed and opaque randomization envelopes. Study interventions were performed in a single-blind design, while participants in the no-treatment control group were necessarily unblinded. All participants provided written informed consent. The study protocol was approved by the ethical committee

**Fig 1.** Study design (a) and experimental procedure (b).

of the Medical Faculty at Ludwig-Maximilians-University Munich (no. 402–13) and was registered retrospectively at the German Clinical Trials Register (no. DRKS00015192).

## Participants

Inclusion criteria comprised age between 18 and 50 years, normal body weight and normal or corrected-to-normal vision and hearing. Exclusion criteria were metal implants or implanted device, presence of acute or chronic disease, and regular intake of drugs (except for hormonal contraceptives, thyroid medications, and allergy medications). Furthermore, volunteers were excluded when they presented with anxiety and depression scores above the clinically relevant cut-off score according to the 'Hospital Anxiety and Depression Scale' [HADS; 13], when they scored lower than 80 in the 'Motion Sickness Susceptibility Questionnaire' [MSSQ; 14], or when they developed less than moderate nausea (<5 on a 11-point numeric rating scale (NRS), with '0' indicating 'no nausea" and '10' indicating 'maximal tolerable nausea') during a 20-min exposure to the nauseating vection stimulus on a pretest day. Placebo and no treatment groups were comparable at baseline with regard to sociodemographic and clinical characteristics (Table 1).

## Experimental procedure

The experimental procedure is summarized in Fig 1B. Each participant was tested on two separate days at least 24 hours apart at the same daytime between 14.00 and 19.00 h after a fasting

**Table 1. Sample characteristics at baseline.**

|  | No treatment (n = 30) | Placebo treatment (n = 60) | P* |
|---|---|---|---|
| Sex, *m/f* | 15/15 | 30/30 | 1 |
| Age, *mean (s.d.)* | 23.5 (2.7) | 23.5 (3.4) | 0.49 |
| Education ($\geq$ high school degree), *n (%)* | 27 (90) | 59 (98) | 0.58 |
| Nonsmoker, *n (%)* | 26 (87) | 53 (90) | 0.73 |
| Body Mass Index, *mean (s.d.)* | 22.3 (2.7) | 21.6 (2.1) | 0.18 |
| MSSQ, *mean (s.d.)* | 130.3 (38.4) | 137.1 (39.9) | 0.44 |
| HADS-anxiety, *mean (s.d.)* | 4.0 (2.6) | 4.0 (2.2) | 0.95 |
| HADS-depression, *mean (s.d.)* | 1.4 (1.6) | 1.8 (1.6) | 0.35 |
| STAI-state anxiety, *mean v* | 34.7 (4.9) | 35.8 (8.3) | 0.51 |
| STAI-trait anxiety, *mean v* | 38.8 (6.4) | 37.8 (6.5) | 0.51 |
| PSQ-Stress, *mean v* | 31.1 (14.3) | 30.0 (16.1) | 0.72 |

Abbreviations: MSSQ, Motion Sickness Susceptibility Questionnaire; HADS, Hospital Anxiety and Depression Scale; STAI, State-Trait Anxiety Inventory; PSQ, Perceived Stress Questionnaire.

*Results of χ2 tests, Kruskal-Wallis tests, or univariate ANOVA, as appropriate.

period of at least 3 h. The session on Day 1 started with a 20-min resting period and then the vection stimulus was turned on for 20 min. On Day 2 after a 10-min resting period, participants were randomly allocated to treatment or no treatment. For participants in the treatment groups, a standardized expectancy manipulation procedure was performed and the assigned treatment was turned on for 20 minutes. Participants in the no treatment groups remained untreated. The sessions on both testing days ended with a 20-min recovery period. On both days, the electrogastrogram, the electrocardiogram, respiration frequency, and the electroencephalogram (including electrooculogram) were recorded, and subjects rated the intensity of perceived nausea and other symptoms of motion sickness (MS). Plasma samples for proteomics assessments and saliva samples for cortisol measurement (results not reported here) were collected during baseline and at the end of the vection stimulus.

## Interventions

Placebo and active interventions were implemented by means of a programmable transcutaneous electrical nerve stimulation (TENS) device (Digital EMS/TENS unit SEM 42, Sanitas, Uttenweiler, Germany). For the active intervention, the electrodes were placed around 'PC6', a validated acupuncture point for the treatment of nausea [15, 16], and the TENS program was turned on for 20 minutes. For the placebo treatment the electrodes were attached just proximal and distal to a non-acupuncture point at the ulnar side of the forearm generally accepted to represent a dummy point in the context of acupuncture research [17]. Two types of placebo stimulation were applied: 30 participants (15 males, 15 females) received subtle stimulation at a very low intensity by turning on the massage program of the TENS device, while 30 participants (15 males, 15 females) received no electric stimulation at all. Since the two placebo interventions reduced nausea and MS to a similar extent [11], participants from both groups were combined in the present analyses into one placebo group (n = 60).

## Nausea induction

Nausea was induced by standardized visual presentation of alternating black and white stripes with left-to-right circular motion at 60 degree/sec. This left-to-right horizontal translation induces a circular vection sensation wherein subjects experience a false sensation of translating to the left [18, 19]. The nauseating stimulus was projected to a semi-cylindrical and semi-transparent screen placed around the volunteer at a distance of 30 cm to the eyes. Such stimulation simulates visual input provided by a rotating optokinetic drum, commonly used to induce vection (illusory self-motion) and thereby nausea [20, 21]. For security reasons, the vection stimulus was stopped if nausea ratings indicated severe nausea (ratings of 9 or 10 on the 11-point NRS).

## Behavioral and psychophysiological measurements

Perceived nausea intensities were rated at baseline and every minute during the nausea period on 11-point NRSs, with '0" indicating 'no nausea" and '10" indicating 'maximal tolerable nausea". Symptoms of MS were assessed by using the 'Subjective Symptoms of Motion Sickness' (SSMS) questionnaire [adapted from 14], with scores of 0 to 3 assigned to responses of none, slight, moderate, and severe for symptoms of dizziness, headache, nausea/urge to vomit, tiredness, sweating, and stomach awareness, respectively.

The electrogastrogram signal, respiratory activity (to control the electrogastrogram for respiratory artifacts), and the electrocardiogram signal (results not reported here) were recorded using a BIOPAC MP 150 device (BIOPAC Systems Inc., Goleta, CA, USA) and Acq-Knowledge4.1 software for data acquisition. The electrogastrogram signal was recorded by

using two Ag/AgCl electrodes (Cleartrace, Conmed, Utica, NY, USA) placed at standard positions on the skin above the abdomen [22]. The skin was cleaned with sandy skin-prep jelly to reduce skin impedance prior to electrode placement (Nuprep, Weaver & Co., Aurora, CO, USA). The electrodes were connected to the BIOPAC amplifier module EGG100C, the signal was digitized at a rate of 15.625 samples per second and filtered using an analog bandpass filter consisting of a 1 Hz first-order low-pass filter and a 5 MHz third-order high-pass filter. Spectral analysis was performed on the last 300 sec of the baseline and nausea periods on each testing day, respectively. Prior to fast Fourier transform, each data epoch was linearly detrended and its ends were tapered to zero using a Hamming window. Spectral power within the normogastric bandwidth (2.5 to 3 cycles per min) and the tachygastric bandwidth (3.75 to 9.75 cycles per min) were determined for three overlapping epochs (Minutes 1 to 3, 2 to 4, 3 to 5) [23]. Finally, the average 'normo-to-tachy ratio' (NTT) was computed as the mean ratio of normogastric to tachygastric spectral power in the three 1-min epochs. NTT is known to decrease during visually-induced nausea, indicating enhanced tachygastric myoelectrical activity and/or reduced normogastric myoelectrical activity [24–26]. NTT data were logarithmized before statistical analysis to obtain approximately normal distributions.

Electrooculography was recorded as part of a 32-lead electroencephalogram (results not reported here) to control for participant´s eye movements during baseline and vection stimulation in order to assure that they followed the standardized instructions, namely to look straight ahead during baseline and to naturally follow the left-to-right horizontal translation of black and white stripes without moving the head during exposure to the vection stimulus, respectively. Horizontal and vertical electrooculography was assessed using the ActiveTwo system (BioSemi, Amsterdam, The Netherlands).

## Proteomic analysis

Blood samples for proteomics assessments were collected in 2.7 ml EDTA tubes (S-Monovette, Sarstedt, Germany) from the antecubital veins and spinned in a centrifuge at 4˚C for 10 min at 3,000g. Plasma samples à 100 µl were stored in 0.5 ml Protein LoBind Tubes (Eppendorf, Germany) at -70˚C until proteomic analysis.

Plasma samples were proteolysed using PreOmics' iST Kit (PreOmics GmbH, Martinsried, Germany) according to manufacturers' specifications. Briefly, undepleted plasma was reduced and alkylated and incubated for 3 hours at 37˚C with Lys-C and trypsin. Resulting peptides were dried for short term storage at -80˚C. Prior to measurement, peptides were resuspended in 2% acetonitrile and 0.5% trifluoroacetic acid. The High Resolution Melt (HRM) Calibration Kit (Biognosys, Schlieren, Switzerland) was added to all samples according to manufacturer's instructions.

Mass spectrometry data were acquired in data-independent acquisition (DIA) mode on a Q Exactive high field mass spectrometer (Thermo Fisher Scientific, Dreireich, Germany). Per measurement 0.5 µg of peptides were automatically loaded to the online coupled Rapid Separation, High Pressure Liquid Chromatography System (Ultimate 3000, Thermo Fisher Scientific, Dreireich, Germany). A nano trap column was used (300 µm inner diameter × 5 mm, packed with Acclaim PepMap100 C18, 5 µm, 100 Å; LC Packings, Sunnyvale, CA) before separation by reversed-phase chromatography (Acquity UPLC M-Class HSS T3 Column 75µm inner diameter x 250mm, 1.8µm; Waters, Eschborn, Germany) at 40˚C. Peptides were eluted from the column at 250 nl/min using increasing acetonitrile concentration (in 0.1% formic acid) from 3% to 40% over a 45-min gradient.

The HRM DIA method consisted of a survey scan from 300 to 1,500 m/z at 120,000 resolution and an automatic gain control target of 3e6 or 120 msec maximum injection time.

Fragmentation was performed via high-energy collisional dissociation with a target value of 3e6 ions determined with predictive automatic gain control. Precursor peptides were isolated with 17 variable windows spanning from 300 to 1,500 m/z at 30,000 resolution with an automatic gain control target of 3e6 and automatic injection time. The normalized collision energy was 28 and the spectra were recorded in profile type.

## Statistical analysis

Statistics were done using MATLAB R2018b. For all statistical tests, a significance level of $p \leq 0.05$ (two-tailed) was assumed. Assumptions of normality were verified for all outcomes before statistical analysis.

**Nausea measures.** For each behavioral and physiological variable (nausea, MS, NTT), day-adjusted scores (DAS) were computed prior to statistical analyses. DAS were calculated as: DAS = $(m_{22} - m_{21}) - (m_{12} - m_{11})$, where $m_{22}$ is measurement 2 on Day 2, $m_{21}$ is measurement 1 on Day 2 and $m_{12}$ and $m_{11}$ are second and first measurement on Day 1, respectively. DAS for each nausea, MS, and NTT were subjected to separate analyses of variance (ANOVAs), with 'group' (placebo group, control group) and 'sex' (male, female) included as between-subject factors.

**DIA data analysis.** Data analysis of DIA files requires comparison of mass spectra against a tailored spectral library built of preceding data dependent mass spectrometry measurements. We searched our DIA files against an in-house library generated from selected mass spectrometry data encompassing 57 files of plasma and serum preparations, spiked with the HRM Calibration Kit. Data dependent files were analyzed using Proteome Discoverer (Version 2.1, ThermoFisher Scientific). Embedded search engine nodes included Mascot (Version 2.5.1, Matrix Science, London, UK), Byonic (Version 2.0, Proteinmetrics, San Carlos, CA), Sequest-HT, and MSAmanda. Peptide false discovery rates (FDR) for all search engines were calculated using percolator node, and the resulting identifications were filtered to satisfy the 1% peptide level FDR (with the exception of Byonic) and combined in a multi-consensus result file maintaining the 1% FDR threshold. The peptide spectral library was generated in Spectronaut (Version 9, Biognosys) with default settings using the Proteome Discoverer combined result file. Spectronaut was equipped with the Swissprot human database (Release 2016.02, 20165 sequences, www.uniprot.org) with a few spiked proteins (e.g., Biognosys HRM peptide sequences). The final spectral library generated in Spectronaut contained 1,811 protein groups, 10,445 proteotypic peptides and 26,805 peptide-precursors.

**Peptide dataset.** The DIA mass spectrometry data were analyzed using the Spectronaut 9 software applying default settings with the following exceptions: quantification was limited to proteotypic peptides, data filtering was set to Qvalue sparse for the peptide-based analysis. The Qvalue sparse setting includes all observations that pass the Qvalue at least once and it generates a matrix with a minimum of missing values. A peptide dataset used for analyses of covariance (ANCOVA) was generated by removing peptides with >10% missing values. Then data were normalized by setting the median to one, and intensities were log transformed (logN) for further analyses.

**Protein dataset.** For the protein-based analysis, intensities of proteotypic peptides were added up to the respective protein intensities. To ensure best data quality in a most complete protein matrix, data were filled up as follows. Data filtering was set to Qvalue. The Qvalue setting considers only individual observations that pass the Qvalue threshold and generates a matrix containing missing values. To minimize the number of missing values, we used the values generated from the Qvalue sparse setting, representing real mass traces at the respective retention times. Proteins with >5% missing values were deleted. Data were finally normalized

setting the median to one, intensities were log transformed (logN) for further analyses. Fold changes were calculated for both days as log2 fold between measurement 2 and 1.

The mass spectrometry proteomics data have been deposited to the ProteomeXchange Consortium via the PRIDE [27] partner repository with the dataset identifier PXD020563.

**Protein interaction network.** The protein interaction network was created using StringDB Version 1019 using default settings (median confidence). Edges refer to the interaction sources text-mining, experiments, databases, co-expression, neighborhood and co-occurrence.

**GO enrichment analyses.** GO enrichment analyses were done using GO annotation files from http://www.geneontology.org/, releases/2017-03-11. Analysis was restricted to GO Biological Process only. Significantly enriched GO terms were estimated using a hypergeometric distribution test with the full Proteomics Spectronaut 9 Database (1470 unique protein IDs) as background. Fully redundant terms were removed from the list. To form representative functional clusters, similar terms were combined using the Jaccard index $J\_ij = (g\_i \cap g\_j)/(g\_i \cup g\_j)$, where gi and gj are the gene-products assigned to significant enriched pairs of GO-terms i and j, respectively. GO-terms with an index Jij > 0.4 were grouped.

**Dissection of protein variance by experimental factors.** To estimate the variance composition of the protein changes on Day 2 dependent on the three DASs, we used a linear regression model: $y_{p,NS} = \mu + \beta_1 * DAS + \beta_2 * grp + \beta_3 * sex + \beta_4 * DAS * grp + \beta_5 * DAS * sex + \beta_6 * grp * sex + \beta_7 * DAS * grp * sex$, with DASs as the predictor variables and group (grp) and sex as categorical co-variables. To deal with remaining missing data we used multiple imputation to create 5 complete datasets. Missing values were imputed by predictive mean matching using the R package 'mice' (rundata, m = 5, maxit = 100,000, meth = 'pmm', seed = 50, pred = predMat). We used 'bisquare' weight function to detect and remove outliers from the model. The proportion of total variance explained by the regression model was estimated by comparing the regression sum of squares to total sum of squares. Variance composition was computed separately for each imputed dataset. Results were summarized by taking the medians for each protein.

**Placebo-associated protein changes.** *To estimate the difference in placebo-induced protein abundances on Day 2 between* groups, we applied ANCOVA and adjusted for individual differences in protein fold changes during the baseline experiment on Day 1. To further increase statistical power, ANCOVA was performed on the log ratios (measurement 2 vs measurement 1) of peptides instead of protein data. Thus, for large individual proteins we could increase the sample size up to 1,000 fold (S1 Dataset). For each GO term enriched for the ANCOVA-identified proteins we selected all related proteins that were also significantly regulated and generated two linear regression models to predict the three DAS (Nausea, NTT, MS) for the control and placebo groups. A 'bisquare' weight function was used to remove outliers from the model.

**Prediction of placebo responders.** Prediction of placebo responders was based on protein baseline data on Day 2. Only participants were included for which a full proteomics dataset was existent after pre-processing. A one-way ANOVA on protein level was performed to pre-select proteins at baseline of Day 2 that were expressed differentially between placebo responders and placebo non-responders for nausea and MS, respectively. Placebo responders were defined as participants in the placebo groups showing ≥50% reduction in nausea/MS from Day 1 to Day 2. Initially only top 5 proteins based on the F-statistic were selected. Subsequently, we performed sequential feature selection to identify additional proteins with potential to predict good responders. We finally used a linear support vector machine to generate two predictive models for each of the nausea scores. The first model included ANOVA and predictor proteins from sequential feature selection ('ANOVA plus model'). As a null model

reference, we used a model based on randomly selected proteins ('RANDOM model'). Median receiver operating characteristics (ROC) curves and mean area under the curve (AUC) estimation were done using k-fold cross validation with k = 10, with 10 independent permutations.

## Results

### Nausea measures

The placebo-exposed individuals developed fewer symptoms of nausea and MS on Day 2 (Fig 2A and 2B, S1 Table). Consistent with this observation, the mixed-design ANOVA revealed a significantly larger reduction in DAS-Nausea from Day 1 to Day 2 in the placebo group as compared to the control group ($F_{group}(1,86) = 44.83$, $p < 0.001$), with no difference between sexes ($F_{int}(1,86) = 1.01$, $p = 0.32$). After removal of one outlier (Fig 2A) the main effect of 'group' remained significant ($p < 0.001$). Similarly, the ANOVA for DAS-MS indicated relief from MS in the placebo group as compared to the control group ($F_{group}(1,84) = 14.93$, $p < 0.001$), with no difference between sexes ($F_{int}(1,84) = 0.47$, $p = 0.49$). After removal of six outliers (Fig 2B) the main effect of 'group' remained significant ($p < 0.001$).

Thirty-nine out of 60 participants (65%) in the placebo group showed a reduction of at least 50% in DAS-Nausea ('placebo responders'), while only 5 out of 30 (17%) did so in the control group ('non-responders'); the difference between groups was significant ($\chi^2 = 18.69$, $p < 001$). Similarly, 30 out of 58 participants (52%) in the placebo group showed a reduction in DAS-MS of at least 50% compared to 3 out of 29 (10%) in the control group ($\chi^2 = 14.06$, $p < 0.001$).

The ANOVA for DAS-NTT revealed a significant interaction between 'group' and 'sex' ($F_{int}(1,83) = 4.16$, $p < 0.05$; Fig 2C). Contrast analyses indicated a significant difference among treatment groups for female participants ($F_{group}(1,40) = 4.10$, $p < 0.05$), but not for males ($F_{group}(1,43) = 0.83$, $p = 0.40$). After removal of four outliers (Fig 2C) the interaction between 'group' and 'sex' remained significant ($p < 0.05$).

### Proteomic analysis

**Plasma proteome.** We identified 2 samples as haemolytic, which were removed from all analyses. A mass spectrometry-based proteomics approach on peripheral plasma identified 711 proteins represented by 3.224 peptides and 14.588 peptide-precursors, with many proteins showing a marked sex difference in abundance, but no such influence of age, group, day, or time point of measurement (Fig 3A).

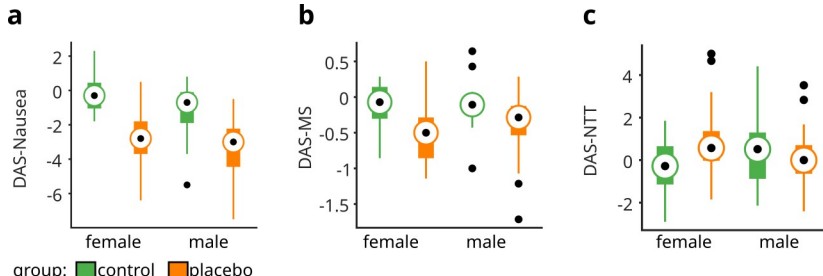

**Fig 2. Placebo effects on nausea ratings, motion sickness, and normo-to-tachy ratio in the electrogastrogram.** Boxplots depict day-adjusted scores (DAS) for the three nausea measures: (a) Nausea (DAS-Nausea), (b) motion sickness (DAS-MS), and (c) normo-to-tachy ratio in the electrogastrogram (DAS-NTT). DAS-Nausea, $F_{group}(1,86) = 44.83$, $p < 0.001$; DAS-MS, $F_{group}(1,84) = 14.93$, $p < 0.001$; DAS-NTT, $F_{int}(1,83) = 4.16$, $p = 0.044$ (female, $F_{group}(1,40) = 4.10$, $p = 0.049$; male, $F_{group}(1,43) = 0.83$, $p = 0.366$); See S1 Table for further details. On each box, the central dot indicates the median, and the bottom and top edges of the box indicate the 25th and 75th percentiles, respectively. The whiskers extend to the most extreme data points not considered outliers, and the outliers are plotted as black dots.

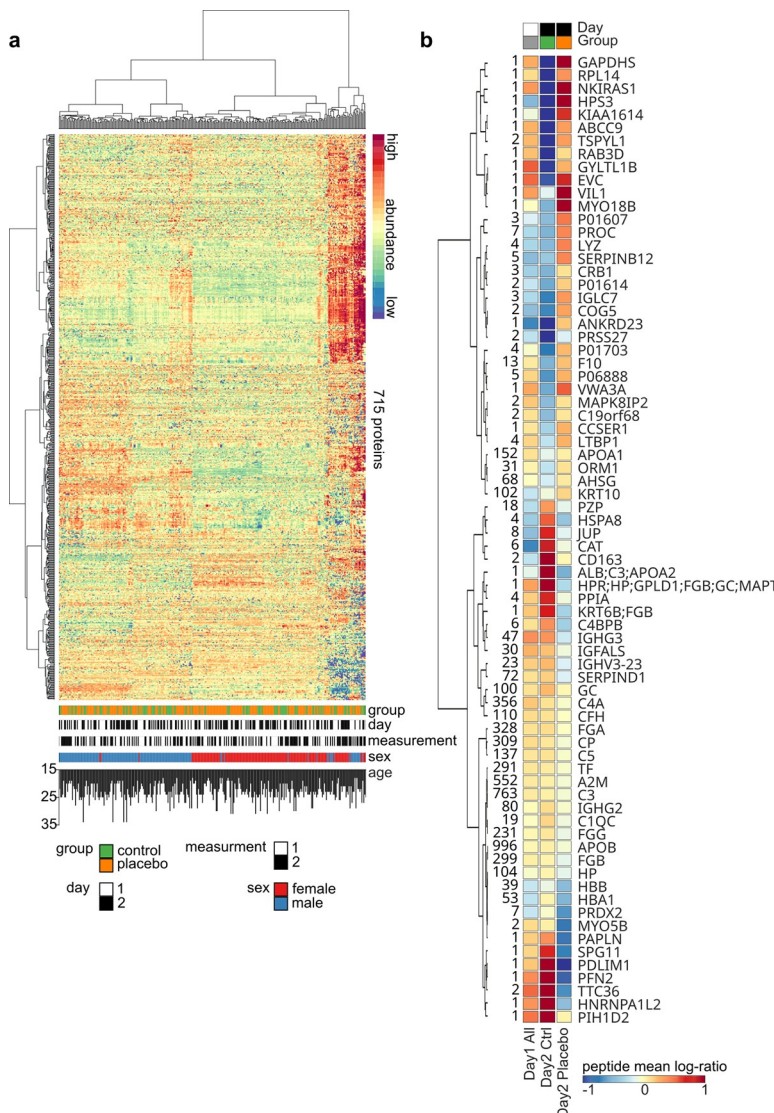

**Fig 3. Identified plasma proteins and those significantly affected by the placebo treatment.** (a) The heatmap depicts the sample (column) vs protein (row) matrix with hierarchical clustering dendrograms. Abundances were row-wise z-score transformed. The sample features group, experiment day, time point of measurement, sex and age are color-/bar-coded at the bottom of the heatmap. (b) Heatmap of the 74 proteins identified as significantly affected ($p < 0.05$) by placebo treatment using ANCOVA, with fold changes on Day 1 included as covariates. Numbers on the left indicate numbers of peptides associated with the protein. Color refers to average log-ratio (measurement 2 vs measurement 1) of all protein-associated peptides. Columns are labeled with the same color-code as in A. Multiple protein labels arise from non-uniquely mapped peptides.

**Dissection of protein variance by experimental factors.** To explore the relationship between fold changes on Day 2 and experimental factors, we first determined the proteins, for which a significant amount of variance could be explained by the factors 'group', 'sex', 'DAS-Nausea' (or 'DAS-MS', 'DAS-NTT'), and by the interaction terms (Fig 4A; S2–S4 Tables). We then performed GO enrichment analyses separately for each type of nausea score to identify the predominant biological processes, in which these proteins are involved (S5–S7 Tables). One of the most significant hits was the GO term 'grooming behavior' in models including 'DAS-NTT' (Fig 4B, S7 Table) with the key proteins 'neurexin-1' (NRXN1; predictive factor

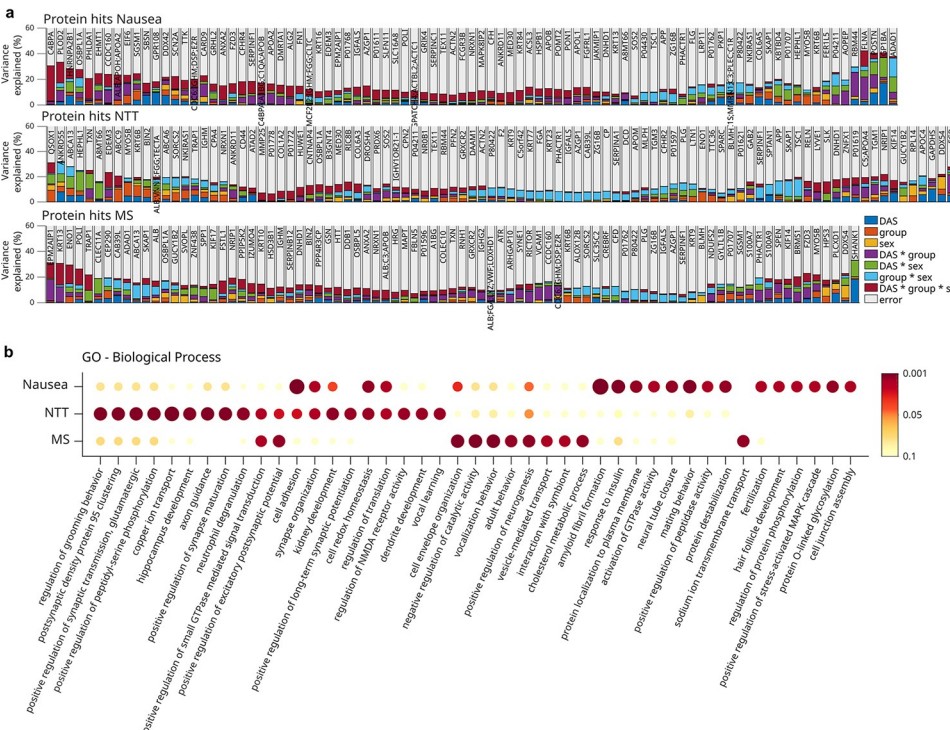

**Fig 4. Variance composition of protein changes on Day 2.** (a) The amount of variance explained was estimated using linear multiple regression models to predict protein fold change variance by different experimental factors (DAS, sex, group) and their interactions (predictor variables). Models were generated independently for each protein and type of nausea measure: Nausea, NTT and MS. Barplot histograms depict variance composition for all proteins significantly affected (p < 0.05) by at least one predictor variable. Multiple protein labels arise from non-uniquely mapped peptides. (b) GO enrichment for each group of significantly regulated proteins (p < 0.05). Dot color and size refer to FDR-corrected enrichment -log10 (p-value).

'group') and 'contactin-associated protein-like 4' (CNTNAP4; 'group * sex'). A further key protein in models including 'DAS-NTT' was reelin (RELN; 'group * sex').

**Placebo-associated protein changes.** To identify proteins affected by the placebo intervention, we next performed ANCOVAs for vection-induced protein fold changes on Day 2 including the factors 'group' and 'sex' as between-subject factors and fold changes on Day 1 as covariate. A significant main effect of 'group' was found for 74 proteins (Fig 3B; S8 Table), indicating differential regulation of these proteins on Day 2 in the placebo group as compared to the control group. In more detail, and relative to the controls, 34 proteins were more abundant following placebo treatment and 30 were less abundant. Mapping these proteins to the StringDB [28] revealed a functional network of 33 proteins as well as 31 unconnected proteins (the remaining 10 proteins could not be mapped uniquely; Fig 5A).

We next performed GO enrichment analyses of these 74 placebo-related proteins and identified 33 enriched GO terms, from which 21 non-redundant terms could be grouped into 8 functional clusters of 37 proteins (FDR-corrected *p* < 0.05; S9 Table). The most striking protein pattern was detected for the GO term 'complement activation', with 8 out of 9 proteins having an equal abundance pattern, namely decrease in the placebo group as compared to the control group (Fig 5B). Involved proteins were complement C3, C4a and C5, complement factor H (CFH), complement C1q subcomponent subunit C (C1QC), immunoglobulin heavy variable 3–23 (IGHV3_23), and immunoglobulin heavy constant gamma 2 and 3 (IGHG2, IGHG3).

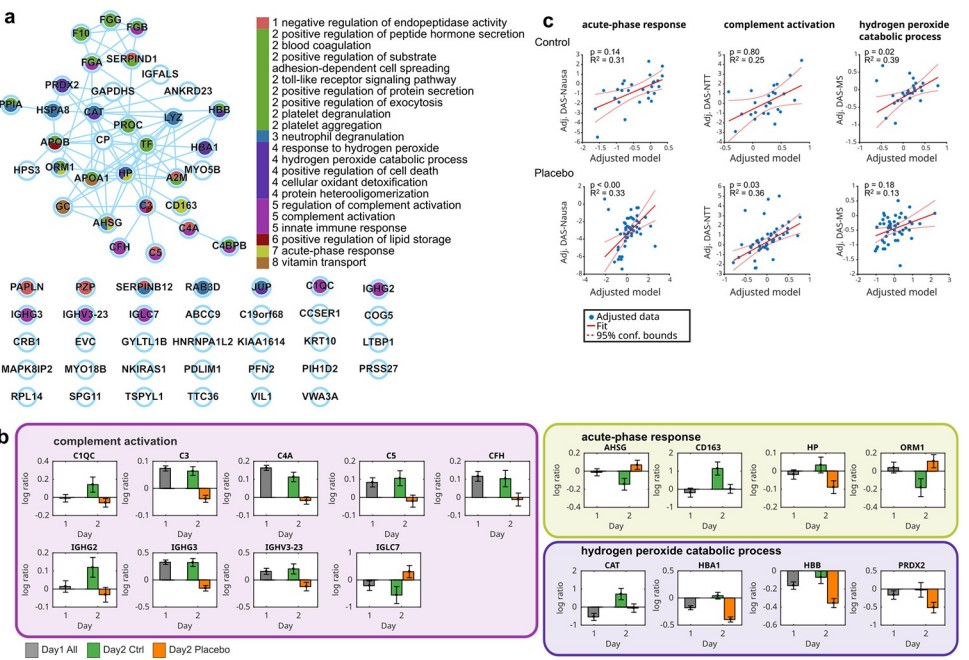

**Fig 5. StringDB network and Gene Ontology (GO) enrichment analysis of placebo-related proteins.** (a) StringDB network of placebo-related proteins. Nodes are colored according to Gene Ontology (GO) term group association. Numbers refer to functional clusters. GO functional clusters were created based on shared members (proteins). Edges refer to StringDB interactions (see methods). (b) Expression patterns for selected GO term proteins. Barplots depict average log expression (measurement 2 versus measurement 1) of all protein-associated peptides. Barplot colors refer to day and group. Error bars denote standard error of the mean. Boxes around barplots indicate GO term membership, and the color of the boxes refer to GO functional cluster associated in panel A. (c) The fitted GO term-based linear multiple regression models to predict DAS. Acute-phase response proteins predict DAS-Nausea in control (top) and placebo group (bottom); complement activation proteins predict DAS-NTT in control and placebo groups; hydrogen peroxide catabolic process proteins predict DAS-MS in control and placebo groups; Model p-values (FDR-corrected) and model R-squared are specified in the plots. Blue dots are model outputs for each data point. The linear fit and 95% confident bands are denoted by solid and dashed red lines.

To evaluate the predictive value of GO term related proteins for nausea indices, we performed regression analyses for each DAS nausea score in the placebo and control groups, respectively (S1 File, S10 Table). In the placebo group, DAS-Nausea was best predicted by fold changes of proteins involved in the acute phase-response, namely haptoglobin precursor (HP), alpha-1-acid glycoprotein 1 (ORM1), alpha-2-HS-glycoprotein (AHSG), and 'scavenger receptor cysteine-rich type 1 protein M130' (CD163). Furthermore, DAS-NTT was predicted by proteins related to the GO terms 'blood coagulation' and 'complement activation' (Fig 5C; S10 Table). Key proteins for the association with blood coagulation were fibrinogen alpha and beta (FBA and FBB; S10 Table). In the control group, DAS-MS was predicted by the GO terms 'hydrogen peroxide catabolic process', 'positive regulation of substrate adhesion-dependent cell spreading', and 'protein heterooligomerization' (Fig 5C; S10 Table). Key proteins for these GO terms were apolipoprotein A-I (APOA1), hemoglobin subunit beta (HBB), and hemoglobin subunit alpha (HBA1).

**Prediction of placebo responders.** We finally aimed to identify Day 2 baseline proteins that could predict placebo responders using support vector machines (see methods). True negative, false negative, false positive and true positive values of the predictive models for nausea and MS are shown in the confusion matrices (Fig 6A). AUC estimators for the 'ANOVA plus model' were 0.86 for nausea and 0.93 for MS, respectively, compared to 0.6 ± 0.07 for the 'RANDOM model' (Fig 6B). Proteins differentiating between placebo responders and nonresponders (S11 and S12

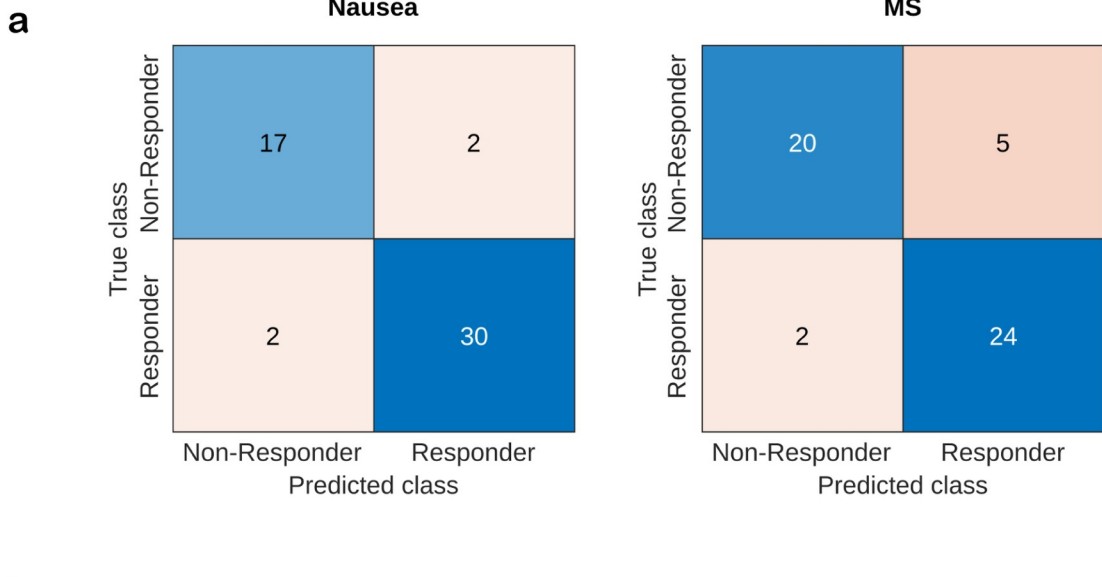

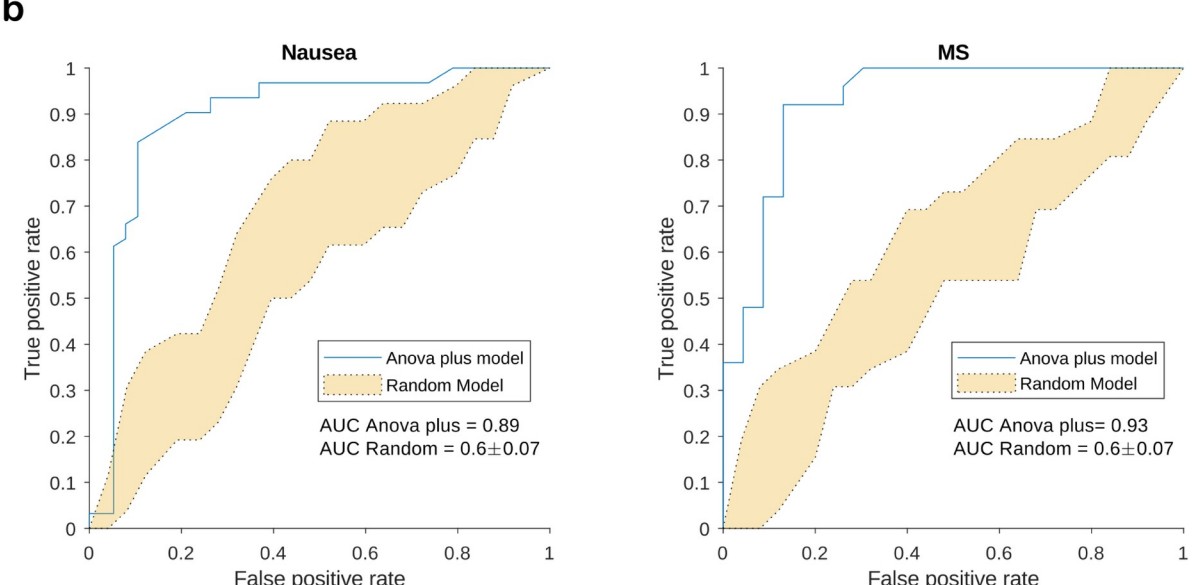

**Fig 6. Prediction of placebo responders by Day 2 baseline proteins.** (a) Adapted confusion matrices for placebo responders vs. non-responders in the placebo group for nausea and motion sickness. Placebo responders were defined as participants showing a reduction of at least 50% in day-adjusted scores for nausea (DAS-Nausea) and motion sickness (DAS-MS), respectively. (b) Receiver operating characteristic (ROC) curves for the support vector machine (SVM) models. The blue line refers to the 'ANOVA plus model'. Yellow area refers to the range of all 10 'RANDOM model' permutations, area under the curve (AUC) values in mean and standard deviation given.

Tables) comprised immunoglobulins (IGHM, IGKV1D-16, IGHV3-23, IGHG1) and MASP2, which are related to regulation of complement activation, as well as proteins related to oxidation reduction processes (QSOX1, CP, TXN). Also included was SLC9A3R1, a protein related to various biological processes, including dopamine receptor signaling.

## Discussion

Moving beyond recent studies, which used genomic techniques to uncover the mechanistic basis of placebo effects [29, 30], we here discovered for the first time specific molecular

signatures reflecting the placebo effect and its impact in acute nausea using a proteomics approach. We identified distinct biological processes that were associated with the placebo effects. For example, the acute-phase proteins HP, ORM1, AHSG, and CD163 were reduced during placebo treatment and these changes were related to the decrease of nausea in the placebo group. The placebo-associated reduction of FBA and FBB, and the strikingly unique pattern regarding decrease of complement cascade proteins, fit well into this scheme because these indicators of microinflammation are known to increase in response to acute stress [31–39]. Indeed, an early study reported reduction of the acute-phase protein C-reactive protein (CRP) in placebo-treated surgery patients as compared to untreated controls and a recent hypothesis paper postulated that placebo effects are mediated by the suppression of the acute-phase response [35]. Similarly, patients with irritable bowel syndrome showed a reduction of the pro-inflammatory protein osteoprotegerin in response to placebo treatment [8]. Our placebo intervention may thus have dampened microinflammatory processes related to acute nausea and MS.

Proteins most closely associated with the placebo intervention when dissecting the variance of protein fold changes on Day 2 were led by neuropeptides that play a key role in social attachment and affiliation, including NRXN1, CNTNAP4, and RELN. The cell adhesion molecules NRXN1 and CNTNAP4 are involved in mirror neuron activity and empathic behavior [40–44], and RELN has been reported to functionally interact with oxytocin and both neuropeptides have been implicated in autism pathophysiology [45, 46]. Furthermore, NRXN1 has recently been identified as a promising peripheral biomarker for brain-related behavior [47]. Our findings are consistent with earlier studies where administration of oxytocin and vasopressin, prior to eliciting a placebo effect, considerably increased the size of placebo effects in healthy volunteers [48, 49]. Most strikingly, one of our main hits in the GO enrichment analysis was 'grooming behavior'. Grooming in various species has been postulated to constitute an important evolutionary trace of the placebo effect in humans [50–52].

Finally, results indicate that plasma proteomics could be groundbreaking for the identification of biomarkers predicting placebo responders in clinical trials. ROC analyses revealed a protein pattern at baseline of Day 2 that allowed differentiating placebo responders from non-responders with surprisingly high accuracy (Fig 6B). This set of proteins comprised immunoglobulins (IGHM, IGKV1D-16, IGHV3-23, IGHG1) and serum proteases (MASP2), both involved in the regulation of complement activation. Interestingly, changes of proteins related to this pathway were also significantly associated with the size of the gastric placebo effect (S7 Table). Also included were proteins related to oxidative stress reduction (QSOX1, CP, TXN). Furthermore, SLC9A3R1 was part of the predictors, a protein involved also in dopamine receptor binding. The dopaminergic system gets activated before and during placebo interventions [53–55]. In addition, catechol-o-methyltransferase-gene *(COMT)* variants as well as personality traits related to the dopaminergic system explained a significant amount of variance of the placebo effect [56, 57].

Our novel discovery of a proteomic fingerprint of placebo effects in peripheral blood offers transformative potential not only for a better understanding of the molecular basis of the placebo effect in different conditions but also for advancing and simplifying certain categories of clinical research in the future. Placebo arms in clinical trials are scientifically necessary for sound academic research, but they can be ethically irresponsible when novel therapeutics, for example in oncology, offer unprecedented and game-changing benefits. One considerable advantage of precision biomarkers based on plasma proteins is that peripheral blood is easily accessible. Once successfully validated across diseases, the inclusion of placebo control groups in clinical trials may no longer be necessary. Furthermore, the success of clinical trials depends on assay sensitivity, i.e., the ability to distinguish an effective treatment from a placebo or

other control treatment [58]. Since large placebo effects decrease assay sensitivity, it is of utmost importance to learn more about the characteristics of good placebo responders. Our results are very promising in this regard.

Some limitations of our study need to be acknowledged. First, identification of low abundant, circulating proteins in plasma using mass spectrometry is limited in sensitivity due to a huge dynamic range of proteins in plasma. To increase reliability of results, however, we did not focus on single protein hits but rather analyzed variations in GO pathways to detect changes in biologically connected patterns. This strategy successfully identified several biological processes, and importantly, these same processes had been shown to be relevant to placebo phenomena [8, 35, 37, 48–57]. Secondly, mass spectrometry-based protein identification is always dependent on algorithms, which are constantly improved (eg by artificial intelligence approaches). Hence our results represent a snap shot of the current state-of-the art during time of analyses. However, since we deposit all raw data to the publicly accessible PRIDE repository, we contribute an important resource exploitable in the future by improved analysis pipelines. Third, we pursued an unbiased global proteomics approach to uncover a blood plasma protein signature imprinted by the placebo effect in humans and provided proof-of-principle evidence that circulating proteins predict and reflect the placebo effect in humans. However, we recognize that our results have to be regarded preliminary as long as validation studies in larger populations are missing.

In conclusion, our results indicate that plasma proteomics is a timely and promising approach to quantify and predict the placebo effect in nausea and to better understand its molecular basis. Future studies are warranted to validate our findings in larger populations and other clinical conditions, such as chronic pain and depression. This could not only foster the understanding of molecular placebo mechanisms underlying across medical conditions, but also help to optimize clinical research methodology.

## Supporting information

**S1 Dataset. Plasma proteins and peptides.**
(XLSX)

**S1 File. Prediction of placebo effects by Gene Enrichment (GO) terms.**
(TXT)

**S1 Table. ANCOVA results for DAS-Nausea, DAS-MS, and DAS-NTT.**
(PDF)

**S2 Table. Proteins for which a significant amount of variance could be explained by 'group', 'sex', 'DAS-Nausea', or by any of the interaction terms.**
(PDF)

**S3 Table. Proteins for which a significant amount of variance could be explained by 'group', 'sex', day-adjusted scores of motion sickness (DAS-MS), or by any of the interaction terms.**
(PDF)

**S4 Table. Proteins for which a significant amount of variance could be explained by 'group', 'sex', day-adjusted scores of normo-to-tachy ratio (DAS-NTT), or by any of the interaction terms.**
(PDF)

**S5 Table. Enriched GO groups based on proteins for which a significant amount of variance could be explained by the factors 'group', 'sex', 'DAS-Nausea', or by any of the interaction terms.**
(PDF)

**S6 Table. Enriched groups of proteins for which a significant amount of variance could be explained by the factors 'group', 'sex', 'DAS-MS', or by any of the interaction terms.**
(PDF)

**S7 Table. Enriched groups of proteins for which a significant amount of variance could be explained by the factors 'group', 'sex', 'DAS-NTT', or by any of the interaction terms.**
(PDF)

**S8 Table. 74 proteins that were differentially regulated in the placebo group as compared to the control group.** Retrieved from analyses of covariance (ANCOVA) on vection-induced fold changes of proteins on Day 2 (between-subject factors 'group' and 'sex', covariate fold changes on Day 1).
(PDF)

**S9 Table. Significant Gene enrichment (GO) terms and related proteins resulting from GO enrichment analyses on 74 proteins differentially regulated in the placebo and control groups.** For list of 74 proteins, see S8 Table.
(PDF)

**S10 Table. Significant prediction of DAS-Nausea, DAS-MS, and DAS-NTT in the placebo and control groups by protein fold changes of significantly enriched GO terms.** For GO terms, see S9 Table.
(PDF)

**S11 Table. Proteins at baseline of Day 2 differentiating between placebo responders (≥50% reduction in nausea) and placebo non-responders according to ROC curves.**
(PDF)

**S12 Table. Proteins at baseline of Day 2 differentiating between placebo responders (≥50% reduction in motion sickness) and placebo non-responders according to ROC curves.**
(PDF)

## Acknowledgments

We would like to thank Franziska Stahlberg and Simone Aichner for their appreciated support in conducting the nausea experiment, and Nicole Senninger and Nicole Eckhard for their excellent technical assistance.

## Author Contributions

**Conceptualization:** Karin Meissner, Christine von Toerne, Stefanie M. Hauck, Matthias H. Tschoep.

**Data curation:** Karin Meissner, Dominik Lutter, Uli Ohmayer.

**Formal analysis:** Karin Meissner, Dominik Lutter, Christine von Toerne, Anja Haile, Stephen C. Woods, Verena Hoffmann, Uli Ohmayer, Stefanie M. Hauck.

**Funding acquisition:** Karin Meissner, Matthias H. Tschoep.

**Investigation:** Anja Haile, Verena Hoffmann.

**Methodology:** Karin Meissner, Dominik Lutter, Christine von Toerne, Stefanie M. Hauck, Matthias H. Tschoep.

**Project administration:** Anja Haile, Verena Hoffmann.

**Supervision:** Karin Meissner, Christine von Toerne, Stefanie M. Hauck, Matthias H. Tschoep.

**Visualization:** Dominik Lutter.

**Writing – original draft:** Karin Meissner, Dominik Lutter, Christine von Toerne, Stefanie M. Hauck.

**Writing – review & editing:** Karin Meissner, Dominik Lutter, Christine von Toerne, Anja Haile, Stephen C. Woods, Verena Hoffmann, Uli Ohmayer, Stefanie M. Hauck, Matthias H. Tschoep.

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
