## [Decision Letter · Decision Letter 0]

16 Jun 2020

PONE-D-20-09037

Molecular classification of the placebo effect in nausea

PLOS ONE

Dear Dr. Meissner,

Thank you for submitting your manuscript to PLOS ONE. After careful consideration, we feel that it has merit but does not fully meet PLOS ONE’s publication criteria as it currently stands. Therefore, we invite you to submit a revised version of the manuscript that addresses the points raised during the review process.

We look forward to receiving your revised manuscript.

Kind regards,

Yi Hu

Academic Editor

PLOS ONE

Journal Requirements:

2. We noted in your submission details that a portion of your manuscript may have been presented or published elsewhere.

"The data was collected as part of a clinical trial, which has been published elsewhere (https://doi.org/10.3389/fnins.2019.01212). None of the proteomics data have been published before, nor are they under consideration for publication elsewhere. "

Reviewers' comments:

Reviewer's Responses to Questions

**Comments to the Author**

1. Is the manuscript technically sound, and do the data support the conclusions?

Reviewer #1: Yes

Reviewer #2: Partly

Reviewer #3: Yes

2. Has the statistical analysis been performed appropriately and rigorously? 

Reviewer #1: Yes

Reviewer #2: Yes

Reviewer #3: Yes

3. Have the authors made all data underlying the findings in their manuscript fully available?

Reviewer #1: Yes

Reviewer #2: Yes

Reviewer #3: Yes

4. Is the manuscript presented in an intelligible fashion and written in standard English?

Reviewer #1: Yes

Reviewer #2: Yes

Reviewer #3: Yes

5. Review Comments to the Author

Reviewer #1: This quite innovative study seeks to determine whether the placebo effect can be tracked and predicted in peripheral blood. It is a randomised study including 90 healthy volunteers and using proteomics techniques to identify biomarkers of placebo effect.

In general, the introduction is well contextualized, the statistics are correct, the results are well presented and the references appropriate and updated.

The following suggestions might improve the quality of the manuscript:

- Further details about plasma processing should be provided, i.e., whether abundant proteins such as albumin and immunoglobulins were depleted to avoid interference, or how protein digestion was performed.

- Qvalue sparse can lead to an accumulation of false positives in the candidates list when the spectral library recovery is poor (< 50%) and the data completeness is low (< 65%). Proteins were filtered if missing values were above 95%. Depending on the distribution of missing values, the use of Qvalue sparse to complete the Qvalue matrix can lead to a false positive accumulation if the % of missing values in the filtered matrix is still high (and this could happen if proteins are kept when presenting 94% of missing values). Thus, what is the data completeness in the Qvalue filtered matrix and the spectral recovery in these samples?

- Significantly enriched GO terms were estimated using a hypergeometric distribution test with the full Proteomics Spectronaut 9 Database (1470 unique protein IDs) as background. In figure 3b only Biological process GO terms are presented. What is the full extent of this analysis? (i.e. were molecular pathways, cellular components or other GO categories also evaluated? Is there any multi-testing correction when estimating GO enrichment significance?).

- Proteins with 95% of missing values seem to be a loose criterion. Review this statement and if it is correct, a more severe criterion needs to be used.

- In the ANOVA tests, was any multi-testing correction applied?

- To estimate the potential utility of these findings, the percentage of volunteers having changes in protein expression and the percentage of volunteers not having them in the different groups should be presented. This information is required to support the hypothesis that the placebo group could be omitted in the future.

Minor comments:

At the end of the introduction, the text following the study objectives should be in the section of patients & methods.

Reviewer #2: Very interesting piece of work on how placebo treatment will induce modification in the plasma proteome.

Some general comments:

There is a misuse of abbreviations throughout the manuscript, that makes the reading hard to follow. This is evident and can led to conclusion in the M&M section. This must be address and include abbreviation correctly.

The introduction is too brief and must be extensively edited as it resembles an abstract more than an intro. This aspect is evident in the paragraph that include the aims of the study. Please refer to either purpose or goal as one aim, or two separate aims. The segment after nausea must be entirely removed, considering that they are methodological aspects that do not belong to the introduction.

Methods

The method section must be improved, I suggest combining the participants and study design in the same section, including table 1 can be included in this section.

Treatment definition is somewhere lost in the interventions section and should be included in a more visible form in the text.

The randomization and blinding are part of the experimental design and must be before treatment allocation.

Reading the methods section there is no mention to how samples where obtained. You probably did not obtain plasma samples, but peripheral blood samples. Please include the protocol por this. How many mLs where obtained? where?

The behavioral and physiological data shoud included before the proteomic analysis, both the methods section and in the results as they confirm the occurrence of nausea.

There are several statistical concepts in the proteomic analysis that although they may explain some adjusting on the data are unnecessary and probably should be included in the statistical analysis section. Please modify. Also, several abbreviations in this section as pointed out before.

The statistical analysis is extensive and well versed. Nonetheless, the information here presents is somehow disorganized into different analysis. I recommend separating the statistical analysis including subheadings for each endpoint analyzed.

Results

Please remove the participant characteristics as they will be included in the method section.

I would include the proteins in peripheral section at the beginning of the placebo proteome and would change that heading to proteomic analysis. In this section, I believe there is an error directing to Fig. 2b?..I did not see that information reflected in that figure.

I believe the Dissection of protein variance by experimental factors and responder analysis although interesting information is out of place in your study as is not stated on your study aims. It should be removed or presented in a different form.

Discussion

Please remove the phrase “By applying next generation computational and bioinformatics approach”.

Some points to address in the discussion:

how fast could plasma reflect changes in the proteome?. This is an important point considering the rapid effect of the placebo and if the proteome could really reflect this effect.

Please discuss the use of your proper library in the proteomic analysis? Is there some limitation to this?

Please discuss your demographic data?. Although no differences are stated among groups in Table 1. Some things to think about: Could there be an effect of education level?. But, especially of interest is the effect of smokers?.. Is there a relationship between those 4 and 7 individuals in the outlier disposition?. Could smokers be prone to nausea?. Does nicotine could have an effect on nausea?

Considering that your limitations state aspects that may limit the relevance of your results, they should probably explain in a more detailed way and how they could have affected your results.

Rewrite your conclusion according to the aims

Reviewer #3: The manuscript under review: "Molecular classification of the placebo effect in nausea" (Manuscript ID: PONE-D-20-09037), by Karin Meissner, Dominik Lutter, Christine von Toerne, Anja Haile, Stephen C. Woods,

Verena Hoffmann, Uli Ohmayer, Stefanie M. Hauck, and Matthias Tschöp.

The above study constitutes a well executed plasma proteomics study to identify potential protein panels that would discriminate between the placebo versus intervention groups. This clinical study is deemed an important one as it will aid with a more protein-level molecular signature on significant placebo effects normally not accounted for in estimating the effects a given treatment at the minimally invasive plasma level.

The use of English is deemed clear and accurate. The clinical design aspects, including the choice of inclusion/exclusion criteria, sample size, randomization and effect size are satisfactory. The DIA proteomics method along with the instrumental analysis resources used for the analyses is deemed state-of-the-art and demonstrated to achieve deep-enough proteome coverage. The proteomics results data processing and statistics was appropriately chosen. The Figure and tables well completed the manuscript text and facilitated the clarity of the message. The references were up-to-date and relevant to the study presented. Only a couple minor comments apply:

1. As the entire repertoire of plasma proteins were subjected to solution phase proteolysis and subsequent DIA-LC-MS analysis, mainly the higher abundant proteins were characterized. This obviously masked the presence of the anticipated lower abundant proteins with potential placebo relevance. Please comment on the feasibility of the present analysis pipeline of this study to other reported methods that also analyzed non-depleted plasma samples (no prior removal of higher abundant Albumin or IgG proteins) that captured a wider spectrum of lower abundant proteins in randomized pharmacologic intervention settings (i.e. Manousopoulou et al, Clin Nutr. 2019).

2. As an extension to the above query, have the authors considered applying DDA based approaches using analogues instrumental resources (QE HF Orbitrap)? If not, please comment on why not.

It is recommended that the present study be published in PLOS ONE as it will surely aid its readership, provided the above minor comments are addressed.

6. PLOS authors have the option to publish the peer review history of their article (what does this mean?). If published, this will include your full peer review and any attached files.

Reviewer #1: No

Reviewer #2: No

Reviewer #3: Yes: Spiros D. Garbis, PhD

Director, Proteome Exploration Laboratory

California Institute of Technology, Pasadena, California, 91123, USA.

---

## [Author Response · Author response to Decision Letter 0]

31 Jul 2020

Response to reviewers’ comments (PONE-D-20-09037)

• We have adapted the manuscript according to the PLOS ONE’s style requirements.

2. We noted in your submission details that a portion of your manuscript may have been presented or published elsewhere.

"The data was collected as part of a clinical trial, which has been published elsewhere (https://doi.org/10.3389/fnins.2019.01212). None of the proteomics data have been published before, nor are they under consideration for publication elsewhere. " Please clarify whether this [conference proceeding or publication] was peer-reviewed and formally published. If this work was previously peer-reviewed and published, in the cover letter please provide the reason that this work does not constitute dual publication and should be included in the current manuscript.

• A related manuscript of this randomized controlled trial has been published in a peer-reviewed journal. The reason why this work does not constitute dual publication is explained in the cover letter. In the manuscript, we now refer explicitly to the previous publication of the behavioral results in the last paragraph of the introduction section.

• We included captions for Supporting Information files at the end of our manuscript and updated the in-text citations accordingly. 

5. Review Comments to the Author

Reviewer #1: This quite innovative study seeks to determine whether the placebo effect can be tracked and predicted in peripheral blood. It is a randomised study including 90 healthy volunteers and using proteomics techniques to identify biomarkers of placebo effect.

In general, the introduction is well contextualized, the statistics are correct, the results are well presented and the references appropriate and updated.

• We thank the reviewer for the positive evaluation of our manuscript.

The following suggestions might improve the quality of the manuscript:

- Further details about plasma processing should be provided, i.e., whether abundant proteins such as albumin and immunoglobulins were depleted to avoid interference, or how protein digestion was performed.

• We now elaborate on plasma processing in the manuscript and have added the following paragraph:

“Plasma samples were proteolysed using PreOmics' iST Kit (PreOmics GmbH, Martinsried, Germany) according to manufacturers' specifications. Briefly, undepleted plasma was reduced and alkylated and incubated for 3 hours at 37°C with Lys-C and trypsin. Resulting peptides were dried for short term storage at -80°C. Prior to measurement, peptides were resuspended in 2% acetonitrile and 0.5% trifluoroacetic acid. The High Resolution Melt Calibration Kit (Biognosys, Schlieren, Switzerland) was added to all samples according to manufacturer's instructions.“

- Qvalue sparse can lead to an accumulation of false positives in the candidates list when the spectral library recovery is poor (< 50%) and the data completeness is low (< 65%). Proteins were filtered if missing values were above 95%. Depending on the distribution of missing values, the use of Qvalue sparse to complete the Qvalue matrix can lead to a false positive accumulation if the % of missing values in the filtered matrix is still high (and this could happen if proteins are kept when presenting 94% of missing values). Thus, what is the data completeness in the Qvalue filtered matrix and the spectral recovery in these samples?

• We agree and apologize for the misleading description of the data matrix generation. We used a peptide-based approach for the ANCOVA tests (Q value sparse) and all peptides with more than 10% missing values were excluded from the analysis. The protein-based approach (used for all analyses except the ANCOVA analysis) was generated by the Q value setting, generating a data matrix which we then imputed with the values from the Q value sparse analysis, because these representing real mass traces at the respective retention times rather than random values. Only proteins with >95% values were kept in this analysis. We now added an extra column in the protein list, which specifies the number of samples which had a measurement that directly passed the Q value setting, to enable assessment of the validity of the respective protein quantification. Further, we now uploaded and provide access to all original data including spectral files, Qvalue and Qvalue sparse peptide and protein lists via the PRIDE data repository with the dataset identifier PXD020563. The data are accessible for reviewers using the following credentials and the link: 

https://www.ebi.ac.uk/pride/help/archive/reviewers

Username: reviewer41218@ebi.ac.uk

Password: GDBEaaxV

• Further, we would like to point out that we did not use the candidate list output from Spectronaut. And even though we did not use it, we re-confirmed with Biognosys that the values regarding spectral library recovery and data completeness are no longer supported, because they found these assumptions not valid. 

- Significantly enriched GO terms were estimated using a hypergeometric distribution test with the full Proteomics Spectronaut 9 Database (1470 unique protein IDs) as background. In figure 3b only Biological process GO terms are presented. What is the full extent of this analysis? (i.e. were molecular pathways, cellular components or other GO categories also evaluated? Is there any multi-testing correction when estimating GO enrichment significance?).

• We thank the reviewer to bring up this point. Generally, enrichment analyses are a powerful tool for classification and functional identification of genetic and proteomic and other omics data, however, these include several drawbacks. First, enrichments are always biased and limited to the Database/Ontology used for the analysis, and second, significance strongly depends on the background used for enrichment statistics. Both are well known and extensively discussed drawbacks. We decided to only use Biological Processes for interpretation of our results. Molecular Function and Cellular Components in this particular case reveal mostly categories that we found hard to interpret in terms of processes associated to placebo effects from human serum samples. Multiple testing correction (following Benjamini Hochberg) was applied in the GO enrichment analyses, were indicated (see Supplemental tables). 

- Proteins with 95% of missing values seem to be a loose criterion. Review this statement and if it is correct, a more severe criterion needs to be used.

• We thank the reviewer for this comment. Indeed, this section was written misleading and was corrected in the revised version: “Proteins with >5% missing values were deleted.” (see also extended comment above)

- In the ANOVA tests, was any multi-testing correction applied?

• We assume this comment refers to the ANOVA based preselection of candidate proteins for the predictive models. In this case no FDR correction was performed, instead the top 5 Proteins (by F-statistic) were selected. The methods section was edited accordingly.

- To estimate the potential utility of these findings, the percentage of volunteers having changes in protein expression and the percentage of volunteers not having them in the different groups should be presented. This information is required to support the hypothesis that the placebo group could be omitted in the future.

• We understand the reviewer's point. However, we believe that our data do not allow this type of evaluation. First we identified a number of 74 Proteins that differ between the placebo and control group using ANCOVA. Per definition of the statistical test, these are considered as normally/Gaussian distributed within one group. Thus, testing for statistical differences of these proteins within one group is counterintuitive to our understanding. Having said that, we agree with the reviewer that this would be an interesting result, however our study was not designed to test for these changes. We collected 4 blood samples per volunteer on the 2 test days, each on a different time point. A significance test on an individual level is therefore not feasible. 

Minor comments:

At the end of the introduction, the text following the study objectives should be in the section of patients & methods.

• We have deleted this part from the introduction section and report the methodological details now only in the methods section. 

Reviewer #2: Very interesting piece of work on how placebo treatment will induce modification in the plasma proteome.

• We thank the reviewer for the positive evaluation of our manuscript.

Some general comments:

There is a misuse of abbreviations throughout the manuscript, that makes the reading hard to follow. This is evident and can led to conclusion in the M&M section. This must be address and include abbreviation correctly.

• We have checked all abbreviations and now use them only when they are stated at least 3 times in the manuscript, or when they are part of proper nouns. 

The introduction is too brief and must be extensively edited as it resembles an abstract more than an intro. This aspect is evident in the paragraph that include the aims of the study. Please refer to either purpose or goal as one aim, or two separate aims. The segment after nausea must be entirely removed, considering that they are methodological aspects that do not belong to the introduction.

• We have revised and extended the introduction according to the reviewer’s suggestion. In more detail, we extended the background of our study with regard to previous placebo research and the characteristics of proteomic approaches. Furthermore, we revised the paragraph on study aims and removed the methodological details from this section. 

Methods

The method section must be improved, I suggest combining the participants and study design in the same section, including table 1 can be included in this section.

• We report the inclusion and exclusion criteria and participant characteristics now directly after the section on the study design and also included table 1 in this section. However, we still use a separate subheading ‘Participants” for ease of reading. 

Treatment definition is somewhere lost in the interventions section and should be included in a more visible form in the text.

• The nature of the treatments is now reported already in the first paragraph of the methods section (‘Study design’), while the details of the interventions can be found in the paragraph ‘Interventions’.

The randomization and blinding are part of the experimental design and must be before treatment allocation.

• We now report randomization and blinding already in the paragraph ‘Study design’. 

Reading the methods section there is no mention to how samples where obtained. You probably did not obtain plasma samples, but peripheral blood samples. Please include the protocol por this. How many mLs where obtained? where?

• We now report the procedure for blood drawings at the beginning of the paragraph “Proteomic analysis” in the methods section:

“Blood samples for proteomics assessments were collected in 2.7 ml EDTA tubes (S-Monovette, Sarstedt, Germany) from the antecubital veins and spinned in a centrifuge at 4°C for 10 min at 3000g. Plasma samples à 100 μl were stored in 0.5 ml Protein LoBind Tubes (Eppendorf, Germany) at -70°C until proteomic analysis.”

The behavioral and physiological data shoud included before the proteomic analysis, both the methods section and in the results as they confirm the occurrence of nausea.

• Behavioral and physiological data are now consistently reported before the proteomic analysis in both, the methods section and results section. 

There are several statistical concepts in the proteomic analysis that although they may explain some adjusting on the data are unnecessary and probably should be included in the statistical analysis section. Please modify. Also, several abbreviations in this section as pointed out before.

• We moved the statistical concepts from the proteomic analysis section to the statistical analysis section. We also reduced the number of abbreviations in the paragraph. We now use abbreviations only when they are mentioned at least 3 times in the text, or within proper nouns. 

The statistical analysis is extensive and well versed. Nonetheless, the information here presents is somehow disorganized into different analysis. I recommend separating the statistical analysis including subheadings for each endpoint analyzed.

• We have introduced the following subheadings according to the reviewer’s suggestion: ‘Nausea measures’, ‘Spectral library’, ‘Peptide dataset’, ‘Protein dataset’, ‘Protein interaction network’, ‘GO enrichment analyses’, ‘Dissection of protein variance by experimental factors’, ‘Placebo-associated protein changes‘, and ‘Prediction of placebo responders’. 

Results

Please remove the participant characteristics as they will be included in the method section.

• We have moved the part on participant characteristics to the methods section.

I would include the proteins in peripheral section at the beginning of the placebo proteome and would change that heading to proteomic analysis. In this section, I believe there is an error directing to Fig. 2b?..I did not see that information reflected in that figure.

• We have moved the heatmap to the results section and corrected the figure numbers. 

I believe the Dissection of protein variance by experimental factors and responder analysis although interesting information is out of place in your study as is not stated on your study aims. It should be removed or presented in a different form.

• We think that the results are very promising for clinical and placebo researchers and would therefore prefer to keep them. However, we present the results now in a different form: 1) The dissection of protein variance has been moved to the beginning of the proteomic analyses, right after the description of the plasma proteome, in order to give the reader an impression of the extent to which the variance of fold changes on Day 2 is related to experimental factors. 2) The question of placebo responders, and whether they can be predicted by ‘omics techniques, is a central one within the context of placebo research. However, we have added a study in the introduction section, which points to the potential of plasma proteins to predict placebo responders (Kokkotou 2010, https://doi.org/10.1111/j.1365-2982.2009.01440.x). In addition, the study’s aim of identifying placebo responders by plasma proteomics is now explicitly stated in the last paragraph of the introduction section (in the previous manuscript version, the aim was only addressed in the abstract). 

Discussion

Please remove the phrase “By applying next generation computational and bioinformatics approach”.

• We have removed the phrase.

Some points to address in the discussion:

how fast could plasma reflect changes in the proteome?. This is an important point considering the rapid effect of the placebo and if the proteome could really reflect this effect.

• Depending on their site of origin, plasma proteins can change within minutes to hours (Anderson & Anderson, 2002, https://pubmed.ncbi.nlm.nih.gov/12488461/). In order to clarify this point, we now present information on the various origins of the plasma proteome and its temporal dynamics already in the introduction section of the manuscript. 

Please discuss the use of your proper library in the proteomic analysis? Is there some limitation to this?

• The library we used was accumulated from different data-dependent acquisition measurements collected in-house from different plasma and serum samples. It is recommended to use such a tailored sample-specific spectral library to minimize false-positive identifications (Shao & Lam 2016, https://doi.org/10.1002/mas.21512). 

Please discuss your demographic data?. Although no differences are stated among groups in Table 1. Some things to think about: Could there be an effect of education level?. But, especially of interest is the effect of smokers?.. Is there a relationship between those 4 and 7 individuals in the outlier disposition?. Could smokers be prone to nausea?. Does nicotine could have an effect on nausea?

• We performed explorative analyses on whether smoking or education level may affect the severity of nausea but could not find any evidence for such an association. Given the small subgroups of non-smokers and participants with lower education, we prefer not to discuss these points within this manuscript due to its focus on the proteomics results.

Considering that your limitations state aspects that may limit the relevance of your results, they should probably explain in a more detailed way and how they could have affected your results.

• We have extended the limitation section according to the reviewer’s suggestion and added the following sentences:

“Secondly, mass spectrometry-based protein identification is always dependent on algorithms, which are constantly improved (eg by artificial intelligence approaches). Hence our results represent a snap shot of the current state-of-the art during time of analysis. However, since we deposit all raw data to the publicly accessible PRIDE repository, we contribute an important resource exploitable in the future by improved analysis pipelines.”

Rewrite your conclusion according to the aims

• We have revised the conclusions:

“In conclusion, our results indicate that plasma proteomics is a timely and promising approach to quantify and predict the placebo effect in nausea and to better understand its molecular basis. Future studies are warranted to validate our findings in larger populations and other clinical conditions, such as chronic pain and depression. This could not only foster the understanding of molecular placebo mechanisms underlying across medical conditions, but also help to optimize clinical research methodology.” 

Reviewer #3: The manuscript under review: "Molecular classification of the placebo effect in nausea" (Manuscript ID: PONE-D-20-09037), by Karin Meissner, Dominik Lutter, Christine von Toerne, Anja Haile, Stephen C. Woods,

Verena Hoffmann, Uli Ohmayer, Stefanie M. Hauck, and Matthias Tschöp.

The above study constitutes a well executed plasma proteomics study to identify potential protein panels that would discriminate between the placebo versus intervention groups. This clinical study is deemed an important one as it will aid with a more protein-level molecular signature on significant placebo effects normally not accounted for in estimating the effects a given treatment at the minimally invasive plasma level.

The use of English is deemed clear and accurate. The clinical design aspects, including the choice of inclusion/exclusion criteria, sample size, randomization and effect size are satisfactory. The DIA proteomics method along with the instrumental analysis resources used for the analyses is deemed state-of-the-art and demonstrated to achieve deep-enough proteome coverage. The proteomics results data processing and statistics was appropriately chosen. The Figure and tables well completed the manuscript text and facilitated the clarity of the message. The references were up-to-date and relevant to the study presented. 

• We thank the reviewer for the positive evaluation of our manuscript.

Only a couple minor comments apply:

1. As the entire repertoire of plasma proteins were subjected to solution phase proteolysis and subsequent DIA-LC-MS analysis, mainly the higher abundant proteins were characterized. This obviously masked the presence of the anticipated lower abundant proteins with potential placebo relevance. Please comment on the feasibility of the present analysis pipeline of this study to other reported methods that also analyzed non-depleted plasma samples (no prior removal of higher abundant Albumin or IgG proteins) that captured a wider spectrum of lower abundant proteins in randomized pharmacologic intervention settings (i.e. Manousopoulou et al, Clin Nutr. 2019).

• Point well taken, and we now carefully compared our data to the excellent study by Manousopoulou et al. However, in contrast to this study where larger amounts of plasma were pooled across probands and sub-fractionated after iTRAQ labelling, we use single-shot measurements of non-pooled, non-depleted plasma samples (332 in total) which explains the different numbers of protein identifications. The typical and state-of-the art analytical depth of single-shot measurements of undepleted plasma samples results in eg 345 protein IDs (Geyer et al., 2016, https://pubmed.ncbi.nlm.nih.gov/28007936/), or up to 580 protein identifications in >25% of samples (using BoxCar, Wewer-Albrechtsen et al., 2019, https://pubmed.ncbi.nlm.nih.gov/30528273/) or up to 342 proteins in >90% of samples using a DIA approach (Liu et al., 2015, https://pubmed.ncbi.nlm.nih.gov/25652787/). All these values match very well to our achieved identifications rates.

2. As an extension to the above query, have the authors considered applying DDA based approaches using analogues instrumental resources (QE HF Orbitrap)? If not, please comment on why not.

• We have not considered applying DDA-MS measurements for several reasons. In our hands, DIA-MS measurements yielded better results with respect to identification and quantification in plasma which is in line with published reports (e.g. Bruderer et al., 2017, https://doi.org/10.1074/mcp.RA117.000314). Next, our study presents data generated from over 300 MS measurements, which was a huge effort and a lot of financial resources were needed, so we decided to use the technically most robust approach established in our lab. Extra measurements would have generated a huge amount of extra costs. Finally, at the time of analysis, Spectronaut was the only software able to handle these amounts of data (0.5 TB) in a reasonable time frame.

It is recommended that the present study be published in PLOS ONE as it will surely aid its readership, provided the above minor comments are addressed.

---

## [Decision Letter · Decision Letter 1]

19 Aug 2020

Molecular classification of the placebo effect in nausea

PONE-D-20-09037R1

Dear Dr. Meissner,

We’re pleased to inform you that your manuscript has been judged scientifically suitable for publication and will be formally accepted for publication once it meets all outstanding technical requirements.

Kind regards,

Yi Hu

Academic Editor

PLOS ONE

Additional Editor Comments (optional):

Reviewers' comments:

Reviewer's Responses to Questions

**Comments to the Author**

1. If the authors have adequately addressed your comments raised in a previous round of review and you feel that this manuscript is now acceptable for publication, you may indicate that here to bypass the “Comments to the Author” section, enter your conflict of interest statement in the “Confidential to Editor” section, and submit your "Accept" recommendation.

Reviewer #1: All comments have been addressed

Reviewer #2: All comments have been addressed

2. Is the manuscript technically sound, and do the data support the conclusions?

Reviewer #1: Yes

Reviewer #2: Yes

3. Has the statistical analysis been performed appropriately and rigorously? 

Reviewer #1: Yes

Reviewer #2: Yes

4. Have the authors made all data underlying the findings in their manuscript fully available?

Reviewer #1: Yes

Reviewer #2: Yes

5. Is the manuscript presented in an intelligible fashion and written in standard English?

Reviewer #1: Yes

Reviewer #2: Yes

6. Review Comments to the Author

Reviewer #1: (No Response)

Reviewer #2: The authors have addressed all my suggestions and therefore I'm ok with the current form of the manuscript.

7. PLOS authors have the option to publish the peer review history of their article (what does this mean?). If published, this will include your full peer review and any attached files.

Reviewer #1: No

Reviewer #2: **Yes: **Hedie A. Bustamante

---

## [Editor Report · Acceptance letter]

28 Aug 2020

PONE-D-20-09037R1 

Molecular classification of the placebo effect in nausea 

Dear Dr. Meissner:

I'm pleased to inform you that your manuscript has been deemed suitable for publication in PLOS ONE. Congratulations! Your manuscript is now with our production department. 

Kind regards, 

on behalf of

Prof. Yi Hu 

Academic Editor

PLOS ONE